# Heat and cold stress increases the risk of paroxysmal supraventricular tachycardia

**Rakesh Jalali**[1☯], **Jerzy Romaszko**[2☯]*, **Ewa Dragańska**[3], **Leszek Gromadziński**[4], **Iwona Cymes**[3], **Janusz Bernard Sokołowski**[5], **Magdalena Poterała**[6], **Leszek Markuszewski**[6], **Anna Maria Romaszko-Wojtowicz**[7], **Anna Jeznach-Steinhagen**[8], **Katarzyna Glińska-Lewczuk**[3]

**1** Department of Emergency Medicine, University of Warmia and Mazury in Olsztyn, Olsztyn, Poland, **2** Department of Family Medicine and Infectious Diseases, School of Medicine, University of Warmia and Mazury in Olsztyn, Olsztyn, Poland, **3** Department of Water Management and Climatology, University of Warmia and Mazury in Olsztyn, Olsztyn, Poland, **4** Department of Cardiology and Internal Medicine, School of Medicine, Collegium Medicum, University of Warmia and Mazury in Olsztyn, Olsztyn, Poland, **5** Clinical Department of Emergency Medicine, Wroclaw Medical University, Wroclaw, Poland, **6** Department of Medicine, Faculty of Medical Sciences and Health Science, Kazimierz Pulaski University of Technology and Humanities in Radom, Radom, Poland, **7** Department of Pulmonology, School of Public Health, Collegium Medicum, University of Warmia and Mazury in Olsztyn, Olsztyn, Poland, **8** Department of Clinical Nutrition, Medical University of Warsaw, Warsaw, Poland

☯ These authors contributed equally to this work.
* jerzy.romaszko@uwm.edu.pl

**Data Availability Statement:** All relevant data are within the manuscript and its Supporting Information files.

## Abstract

Paroxysmal supraventricular tachycardia (PSVT) is a common arrhythmia in adults. Its occurrence depends on the presence of the reentry circuit and the trigger of the paroxysm. Stress, emotional factors, and comorbidities favour the occurrence of such an episode. We hypothesized that the occurrence of PSVT follows extreme thermal episodes. The retrospective analysis was based on the data collected from three hospital emergency departments in Poland (Olsztyn, Radom, and Wroclaw) involving 816 admissions for PSVT in the period of 2016–2021. To test the hypothesis, we applied the Universal Climate Thermal Index (UTCI) to objectively determine exposure to cold or heat stress. The risk (RR) for PSVT increased to 1.37 (p = 0.006) in cold stress and 1.24 (p = 0.05) in heat stress when compared to thermoneutral conditions. The likelihood of PSVT during cold/heat stress is higher in women (RR = 1.59, p< 0.001 and RR = 1.36, p = 0.024, respectively) than in men (RR = 0.64 at p = 0.088 and RR = 0.78, p = 0.083, respectively). The susceptibility for PSVT was even higher in all groups of women after exclusion of perimenopausal group of women, in thermal stress (RR = 1.74, p< 0.001, RR = 1.56, p = 0.029, respectively). Females, particularly at the perimenopausal stage and men irrespective of age were less likely to develop PSVT under thermal stress as compared to thermoneutral conditions. Progress in climate change requires searching for universal methods and tools to monitor relationships between humans and climate. Our paper confirms that the UTCI is the universal tool describing the impact of thermal stress on the human body and its high usefulness in medical researches.

**Funding:** The author(s) received no specific funding for this work.

**Competing interests:** The authors have declared that no competing interests exist.

## Introduction

Clinical practice indicates that paroxysmal supraventricular tachycardia (PSVT) is a rather frequent reason inducing patients to seek medical assistance. However, the assessment of the actual number of PSVT events is quite problematic. The reason is that PSVT, i.e., rapid heartbeat originating in the atria or atrioventricular node, with the frequency of >100/min, in modern cardiology is divided into a number of types (atrial tachycardias; atrioventricular junctional tachycardias; atrioventricular re-entrant tachycardia), whereas it is still traditionally coded according to the ICD 10 as I47.1 [1, 2]. This leads to some information chaos and generates the aforementioned problems in regards to the assessment of the PSVT incidence. The scarcity of available data is noted by the authors of "2019 ESC Guidelines for the management of patients with supraventricular tachycardia" who assess the incidence as 35 per 100 000 person-years [1]. This number was taken from a rather old (1998) publication by Orejarena LA [3]. Based on more recent data as of 2021, Rehorn M et al. determine the incidence at the level of 57.8 per 100 000 person-years [4]. Given this value, the authors assess that 1/300 of USA citizens has experienced PSVT. The scarcity of epidemiological data is related to, at least partially, quite a diverse spectrum of symptoms (ranging from almost asymptomatic cases to the most common symptoms—racing heartbeat, discomfort, chest pain, fainting) and frequent recurrences of the disease [5]. Some patients, who are aware of their clinical condition, tolerate the disease recurrences quite well. Such patients may successfully attempt to control PSVT at home and do not seek assistance in the emergency departments (EDs), hence they are not included in statistical data [6]. In patients with recurrent PSVT, at least some of the paroxysms may be treated via the vagal nerve stimulation by performing the Valsalva maneuver, drinking ice-cold water, or–less frequently on one's own–the carotid sinus massage. Statistical data concerning the incidence of PSVT will most likely change quite significantly once mobile technologies allowing self-diagnostics become commonly available [7, 8]. The risk of PSVT increases with patients' age and it exceeds fivefold the values for younger people in the group of patients over 65. Females are twice as likely to develop PSVT as males, but this difference decreases with age [3].

Climate change observed for several decades has induced researchers to search for tools to monitor the human-climate relationship. In this respect, the employment of several simple biometeorological parameters such as ambient temperature, wind speed or relative humidity is obviously possible; however, the obtained results are often difficult to interpret and, which is a greater problem, dependent on the geographical location of the conducted study. Hence, modern biometeorological research is based on a number of complex indices that *a priori* take into account the aforementioned components [9]. One of such indices is the Universal Climate Thermal Index (UTCI) that we attempt to popularize in medical sciences [10]. This index is derived from an analysis of human thermal balance with the employment of the multi-node mode of heat transfer (Fiala's model) and it allows one to determine thermoneutral conditions as well as heat and cold stress conditions [11, 12]. The calculation of its value takes into account meteorological data (air temperature, vapor pressure, wind speed, mean radiant temperature) and physiological parameters (metabolic heat, albedo of the body surface and clothing, coefficients of human body and clothing emissivity, thermal and evaporative clothing insulation). The UTCI provides information concerning physiological processes that occur in the human organism in specific external environment conditions [13]. UTCI values are expressed in the degree Celsius and represent the body's thermal load, while its output value allows it to be categorized into one of ten classes (Fig 1) [14, 15].

This index has been comprehensively described in the literature, and its applicability in epidemiological studies with a biometeorological context is beyond doubt [14, 16–18].

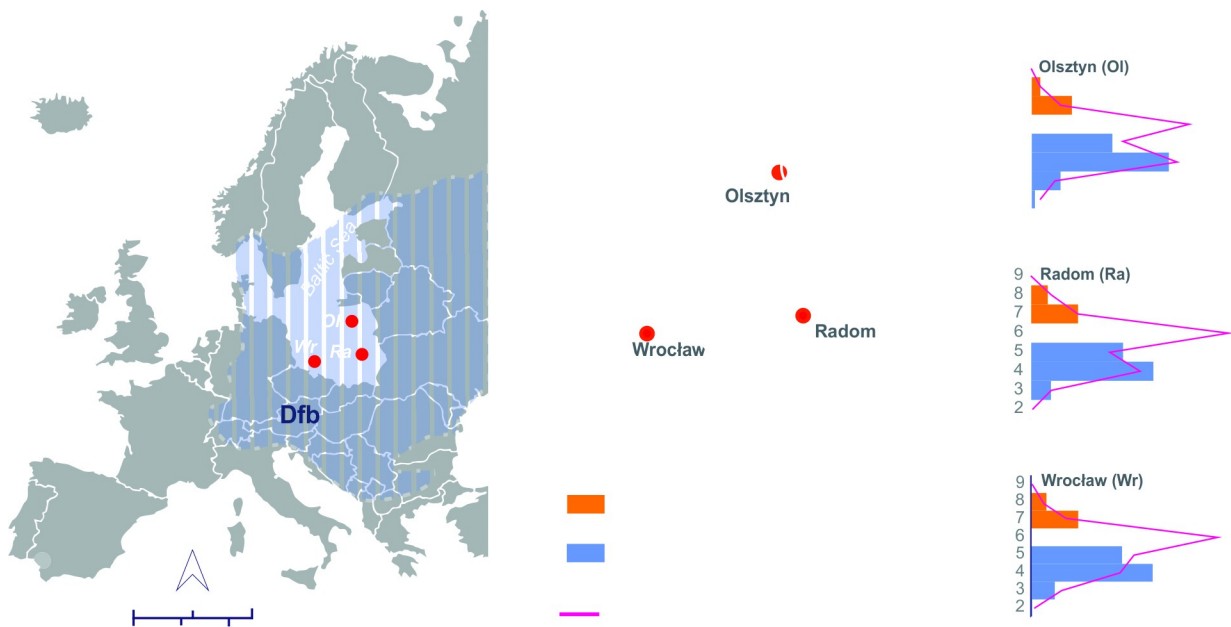

**Fig 1. Location of the study area in relation to Dfb (cold, without dry season with warm summer) climatic zone in Europe.** B–Location of the studied areas on the background of bioclimate types of Poland (Peel et al. 2007). Types of bioclimate of Poland: 1.1.- very small differentiation of thermal clothing insulation throughout the year; slightly load of thermoregulative system due to evaporation 2.1.- small differentiation of thermal clothing insulation throughout the year; very small load of thermoregulative system due to evaporation, 2.2.- small differentiation of thermal clothing insulation throughout the year, small load of thermoregulative system due to respiration; 2.3.- small differentiation of thermal clothing insulation throughout the year; moderate load of thermoregulative system due to respiration, 3.1-moderate differentiation of thermal clothing insulation throughout the year; small load of thermoregulative system due to respiration 3.2.- moderate differentiation of thermal clothing insulation throughout the year; moderate load of thermoregulative system due to respiration, 4.1.- considerable; small load of thermoregulative system due to respiration. Annual mean temperatures for Olsztyn, Radom, and Wrocław for the study period 2016–2021. C—Frequency of UTCI classes and the share of PSVT admissions (females and males) for Olsztyn, Radom, and Wrocław.

Several studies have presented the correlation between the exposure to heat/cold stress and the occurrence of cardiovascular diseases (CVD). For example, whereas, Bhaskaran K. et al., describe the relationship between ambient temperature and the occurrence of myocardial infarctions, Urban A. et al. reported relation between heat/cold stress exposure and CVD mortality and Nguyen J.L. et al. describe dependence of paroxysmal atrial fibrillation on lower temperatures [19–21]. However, to the best of our knowledge, there we were no studies, based on population data, that would either confirm or deny the relationship between the occurrence of PSVT and exposure to heat/cold stress.

The main objective of the study was to determine whether the occurrence of PSVT depends on the biometeorological factors. We also aimed to test the hypothesis that the risk of PSVT is increased by exposure to heat and cold stress. To this end, we tested the usefulness of the UTCI in assessing the susceptibility of different sexes and age groups to PSVT under cold and heat stress.

## Material and methods

The study has been conducted in adherence to STROBE guidelines [22].

### Study population

This is a retrospective analysis of data collected from hospital EDs in three Polish cities–Olsztyn, Radom, and Wrocław covering the period 2016–2021. In total, 816 records in the

**Table 1. Number of patients seeking medical assistance due to PSVT according to the age group and sex in the period from 01.01.2016 to 31.12.2021 in the centers participating in the study.**

| Age group [years] | Total | Females | Males |
|---|---|---|---|
| Total | 816 (100)* | 446 (54.66) | 370 (45.34) |
| 18–35 | 134 (16.43) | 76 (17.04) | 58 (15.68) |
| 36–45 | 126 (15.44) | 84 (18.83) | 42 (11.35) |
| 46–55 | 139 (17.03) | 66 (14.80) | 73 (19.73) |
| 56–65 | 176 (21.57) | 89 (19.96) | 87 (23.51) |
| >65 | 241 (29.53) | 131 (29.,37) | 110 (29.73) |

* numbers in brackets denote the percentage from a given column

healthcare system coded as I.47 according to the ICD 10 (PSVT) were analysed [2]. The majority of patients—54.66% (n = 446) seeking medical assistance due to PSVT were females (Table 1).

To minimize the local character of the study, the occurrences of PSVT were analysed for divergent bioclimatic conditions in Poland, represented by EDs in three Polish cities located in different parts of the country (Fig 1). Olsztyn is a city with a population of about 180,000 inhabitants, situated in north-eastern Poland, Radom–has a population of about 210,000 and is in the south-eastern part of the country, and Wrocław–has about 643,000 inhabitants and is located in south-western Poland.

According to the Köppen climate classification, Poland is characterized by a cold climate without dry season and with warm summer (Dfb), (Fig 1A) [23]. Nevertheless, the variability of climatic conditions between the cities included in the retrospective analysis is noticeable. These differences concern the thermal conditions in particular. In the analysed period, the highest values of the annual average air temperature of 12.9˚C were recorded in south-western Poland (Wrocław), whereas the lowest of 10.6˚C in the north-east (Olsztyn). In Radom the values of this parameter were 11.4˚C (Fig 1B). A similar differentiation level was noticeable for average monthly air temperature. Bioclimatic studies, considering the course of the heat balance of the human body, made it possible to distinguish several types of bioclimates in this relatively small country area (Fig 1B). The cities used for the analysis represent different types of Polish bioclimates according to the classification of Krawczyk B. [24]. Wrocław, with the highest mean air temperature in 2016–2021 among the studied cities, represents the region with low differentiation of thermal clothing insulation over the year, where moderate load of the thermoregulatory system due to respiration occurs (type 2.3). Radom represents the region with moderate differentiation of clothing thermal insulation, where a low load of the thermoregulatory system by respiration occurs (type 3.2). Olsztyn located in the coldest part of the country is characterized by moderate differentiation of thermal insulation of clothing with low or moderate load of the thermoregulatory system due to respiration (type 3.1 and 3.2). The inclusion of these cities to the study allowed for the highest possible, in Polish conditions, climatic differentiation, and, consequently, provided the basis for, at least partial, generalizations as regards the conclusions.

In each of the three cities there are a few hospital EDs that work in the 24/7 mode. We invited one ED from each city to participate in the study. In the analysed period (retrospective analysis) all three EDs, despite the COVID-19 pandemic, provided medical assistance irrespective of the cause of the problem. In the Polish healthcare system, EDs admit patients referred there by their general practitioners and by specialists who provide outpatients care, patients brought by ambulances, as well as those without any referral if their condition requires emergency care. The last group, by definition, embraces patients with symptomatic PSVT.

**Table 2. Frequency (%) of UTCI classes in the analysed cities in the period 2016–2021.**

| Parameter | Cold stress | | | | | Thermo-neutral conditions | Heat stress | | | |
|---|---|---|---|---|---|---|---|---|---|---|
| | Extreme | Very strong | Strong | Moderate | Slight | | Moderate | Strong | Very strong | Extreme |
| UTCI [ºC] | ≤-40.0 | -39.9 to -27.0 | -26.9 to -13.0 | -12.9 to 0.0 | 0.1 to 9.0 | 9.1 to 26.0 | 26.1 to 32.0 | 32.1 to 38.0 | 38.1 to 46.0 | >46.0 |
| Class | 1 | 2 | 3 | 4 | 5 | 6 | 7 | 8 | 9 | 10 |
| City | | | | | | | | | | |
| Olsztyn | n.o. | 0.3 | 6.3 | 29.2 | 17.2 | 36.3 | 8.7 | 2.0 | n.o. | n.o. |
| Radom | n.o. | 0.1 | 4.3 | 25.7 | 19.4 | 36.9 | 9.9 | 3.6 | 0.1 | n.o. |
| Wrocław | n.o. | 0.3 | 5.0 | 25.2 | 18.9 | 37.5 | 9.8 | 3.2 | 0.1 | n.o. |

n.o.–not observed

## Biometeorology

For each studied city, meteorological data were obtained from the Institute of Meteorology and Water Management–State Research Institute (Poland). The meteorological data used in analysis were region–specific and representative for the population under study.

Data were recorded daily at 12.00 UTC (Coordinated Universal Time) and included: air temperature, relative humidity, atmospheric pressure, clouding, wind speed. The meteorological data used in analysis were region–specific and representative for the population under study. Calculation of UTCI values, was performed with the BioKlima2.6 software [25].

Details on the classification and values of the UTCI, with its frequency of occurrence in individual centers are presented in Table 2. Extreme cold and extreme heat classes of UTCI were not observed in any of the cities studied (Table 2). Very strong and strong cold stress occurred most frequently (6.6% days) in Olsztyn, the city representing the coldest region of Poland, while Radom was characterized by the most frequent days (3.7%) of very strong and strong heat stress. In Olsztyn, there were no days with very strong heat stress during the study period. Thermoneutral conditions occurred on average on about 37.2% of days per year.

## Statistical analyses

We performed UTCI-based analysis on the base of appropriate thermal conditions classes and a number of patients with PSVT ascribed to these classes (Fig 1, Table 1) [26]. In order to determine intraannual variations in PSVT for women and men frequencies in consecutive months, where non-integers were used, the HSD Tukey test (one way ANOVA, p<0,05) was applied. In case of analyses where the daily number of patients with PSVT (integers) were ordinal variables and the UTCI classes were employed as a categorical variables we applied the Poisson log-normal test (p<0.05).

Based on UTCI class values, we calculated the relative risks (RR) to quantify the magnitude of the cold (UTCI classes 2–5) and heat stress (UTCI classes 7–9) as factors influencing the incidence rate of PSVT among men and women (outcome). The RR was understood as a ratio of risk outcome with factor present (cold or heat stress) to risk outcome with factor absent (thermoneutral conditions, no thermal stress was considered as a control group) with the 95% lower and upper limits of confidence intervals (CI) [27].

The results were reported for men and women and divided into age groups, separately. The lag effects of biometeorological conditions (UTCI) on PSVT for each age group were estimated using correlation coefficients matrix (Pearson). UTCI lag effect on PSVT was estimated for 1–7 days. The maximum lag of UTCI was set as 7 days in order to capture all thermal stress effects.

Significance of differences before and during the SARS-COV-19 pandemic outbreak (01.03.2020) was tested with the nonparametric ANOVA Mann-Whitney U-test (p>0.05). Statistical analyses were performed with STATISTICA version 13.3 for Windows (TIBCO Software Inc. (2017).

The data set used in the study is presented in S1 Table.

### Ethic statement

This is a retrospective epidemiological study concerning medical events registered routinely in the healthcare system in Poland. Data necessary for the analysis were obtained with the consent of the heads of the respective medical institutions (data were anonymous) and provided by them. No identifiable data were available to the researchers. The Bioethics Committee of the Warmia and Mazury Regional Physicians' Medical Chamber in Olsztyn, to which main authors are affiliated, confirmed on 16 Sep 2021 that this study did not require special consent of the EC (16.09.2021 WMIL-KB/159/2021). Data were accessed for research purposes in January—March 2022 (multicenter study).

## Results

The analysis of the seasonal variability of PSVT cases in consecutive months, revealed no similar trends of PSVT episodes for females and males (Fig 2). Average monthly number of PSVT admissions showed statistically significant differences at p<0.05 for females. Probability of females' admissions with PSVT in April was the lowest (3%), while the highest (10%) during September, a month of strong heat stress. The differences were statistically significant between the two months (p<0.05).

The monthly course of the number (totals) of admissions due to PSVT varies between the age groups of women and men (Fig 3). In the group of men >65, it intensifies in late autumn (November), while in women the frequency of admissions increases significantly in winter and summer during extreme conditions of heat and cold stress. In spring, especially in April, the number of admissions of women with PSVT significantly decreases, regardless of age group.

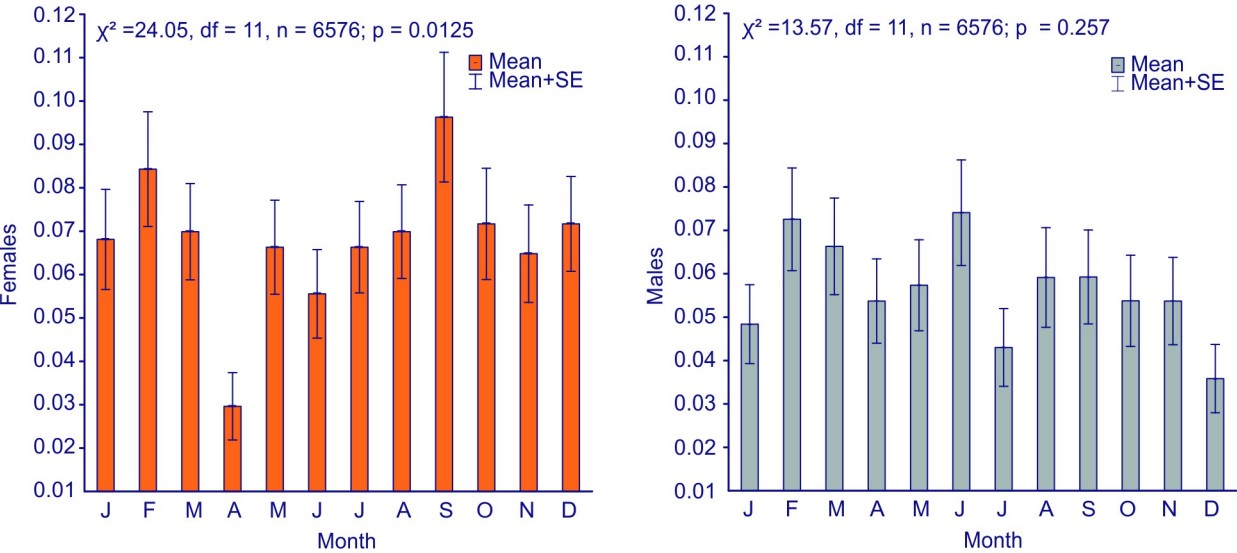

**Fig 2.** Average monthly number of PSVT admissions of females (left) and males (right) in the period of 2016–2021 in the selected cities of Poland (Olsztyn, Wrocław, and Radom,). The differences between mean number of monthly admissions were tested with HSD Tukey test, p<0.05.

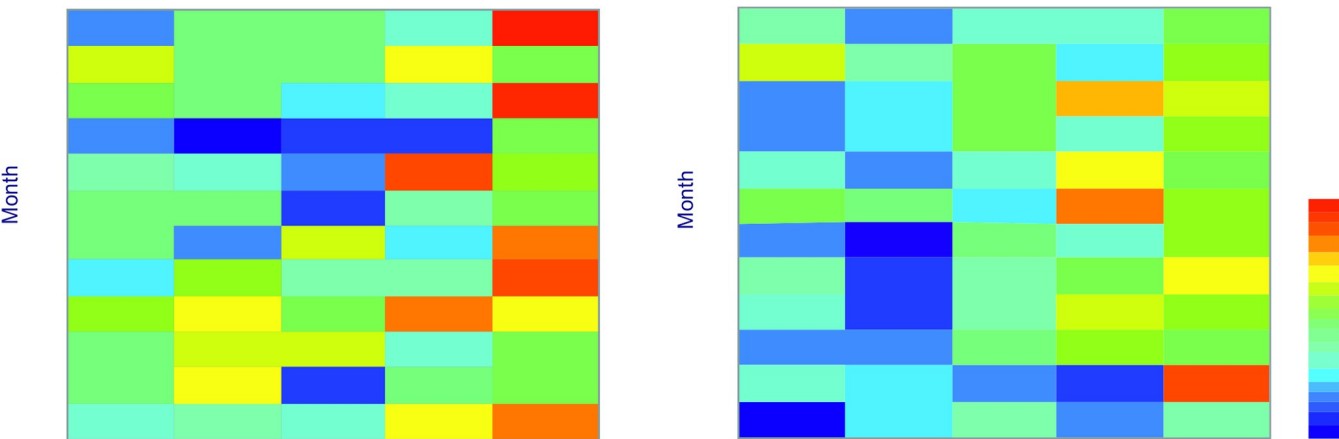

**Fig 3.** Annual variability of PSVT sums for females (left) and males (right). The color scale is relevant to number of admissions with PSVT in a month within an age group.

The analysis of the correlation coefficients between the UTCI values and the daily number of PSVT cases yielded no statistically significant results (p<0.05). Correlations, even if statistically significant, were very weak.

Comparison of the relative risks (RR) for the occurrence of PSVT on days representing strong and very strong cold stress (UTCI classes 2 and 3) or on days with strong and very strong heat stress (UTCI classes 8 and 9) (Table 2) revealed an increased risk compared with thermoneutral conditions (class 6).This outcome, when analysing the data irrespective of age groups (Table 3), is generated by females. Among males, we observed an inverse relationship.

**Table 3. Relative risks (RRs) of PSVT under cold and heat stress conditions (see also S2 Table).**

| Age group | Thermal stress | Females and Males | | | | Females | | | | Males | | | |
|---|---|---|---|---|---|---|---|---|---|---|---|---|---|
| | | RR | 95% CI Lower Limit | 95% CI Upper Limit | P | RR | Lower 95% CI | Upper 95% CI | P | RR | 95% CI Lower Limit | 95% CI Upper Limit | P |
| **All ages** | cold | 1,37 | 1.091 | 1.725 | 0.006 | 1.59 | 1.235 | 2.050 | <0.001 | 0.64 | 0.312 | 1.073 | 0.008 |
| | | | | | | (1.74) * | 1.316 | 2.296 | <0.001 | | | | |
| | hot | 1,24 | 1.002 | 1.576 | 0.050 | 1.36 | 1.041 | 1.771 | 0.024 | 0.78 | 0.417 | 1.350 | 0.083 |
| | | | | | | (1.56) * | 1.174 | 2.082 | 0,002 | | | | |
| **18–35** | cold | 1.59 | 1.077 | 2.403 | 0.020 | 2.13 | 1.217 | 3.720 | 0.008 | 1.01 | 0.554 | 1.806 | 0.999 |
| | hot | 0.44 | 0.223 | 0.845 | 0.014 | 0.85 | 0.407 | 1.778 | 0.667 | no cases | - | - | - |
| **36–45** | cold | 1.50 | 0.891 | 2.499 | 0.128 | 1.46 | 0.750 | 2.675 | 0.282 | 1.60 | 0.711 | 3.600 | 0.256 |
| | hot | 1.50 | 0.891 | 2.499 | 0.128 | 2.20 | 1.195 | 3.771 | 0.011 | no cases | - | - | - |
| **46–55** | cold | 0.60 | 0.343 | 1.042 | 0.069 | 0.94 | 0.489 | 1.719 | 0.786 | 0.24 | 0.078 | 0.7971 | 0.019 |
| | hot | 0.86 | 0.531 | 1.374 | 0.516 | 0.42 | 0.169 | 0.927 | 0.048 | 1.32 | 0.737 | 2.268 | 0.370 |
| **56–65** | cold | 0.95 | 0.630 | 1.399 | 0.758 | 1.64 | 1.055 | 2.825 | 0.042 | 0.46 | 0.212 | 0.8753 | 0.019 |
| | hot | 1.12 | 0.607 | 1.361 | 0.643 | 1.09 | 0.577 | 1.989 | 0.827 | 0.81 | 0.477 | 1.369 | 0.428 |
| **over 65** | cold | 1.15 | 0.791 | 1.654 | 0.474 | 1.75 | 1.061 | 2.863 | 0.028 | 0.59 | 0.312 | 1.073 | 0.085 |
| | hot | 1.38 | 1.174 | 1.940 | 0.049 | 2.00 | 1.238 | 3.230 | 0.004 | 0.80 | 0.462 | 1.350 | 0.387 |

* Calculated with the exclusion of females 46–55 years old. Statistically significant values were marked in bold.

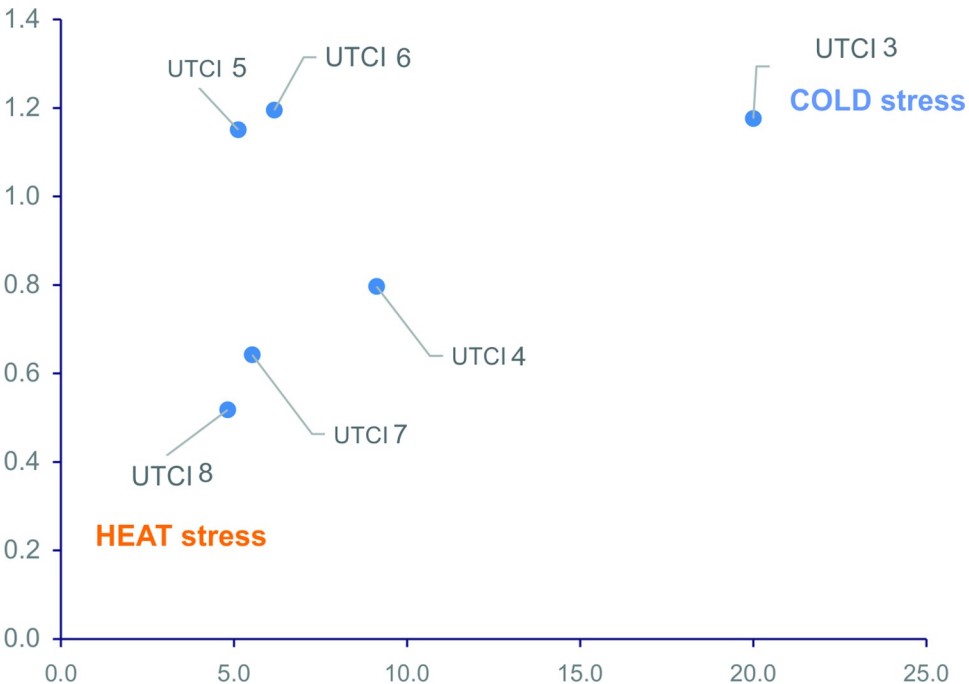

**Fig 4. Response of females at the menopausal stage (age group 46–55) and females in other age groups in relation to cold and heat stress.**

By dividing patients into age groups, a priori, we hoped for an "interesting" result in the group of females 46–55 years old (perimenopause stage) [28]. In this age group (and only in this one) the RR of PSVT for females is lower in thermoneutral conditions than in cold/heat stress conditions. This result is similar to the ones obtained for males (Fig 4).

Interestingly, the exclusion of females in the age group of 46–55 years from the calculation of the RR presented in Table 3 would lead to the increase of the RR for females in heat stress conditions up to 1.59.

Our study is based on data from three cities in different geographical locations in Poland, which have different biometeorological conditions (Fig 1A and 1B). The distribution of PSVT frequencies on days determined by UTCI classes at each location studied is similar (Fig 1C). The relative risks, both overall and subdivided by age groups, calculated separately for each city, give similar values (S2 Table), which allows generalization of the results obtained to the aggregated values presented in Table 3.

The analysis also accounted for the possibility of the delayed effect (lag time) from 1 to 7 days. The linearity of the curves representing the cumulative number of PSVT cases in females and males in heat and cold stress conditions indicate the absence of the cumulative effect (Fig 5).

It is noteworthy that the course of the two cumulative curves (hot and cold) is almost parallel in males and females (no temporal clustering). However, the group of females clearly confirms a greater sensitivity to heat and cold stress than that of males.

Since our retrospective analysis involved data from the periods both before and during the COVID-19 pandemic, we also examined the potential impact of that pandemic on the obtained results. We checked effects of COVID-19 on the number of daily admissions due to PSVT per 1 day in the study period and RRs—it was statistically insignificant at p<0.05 (not shown in the tables or figures).

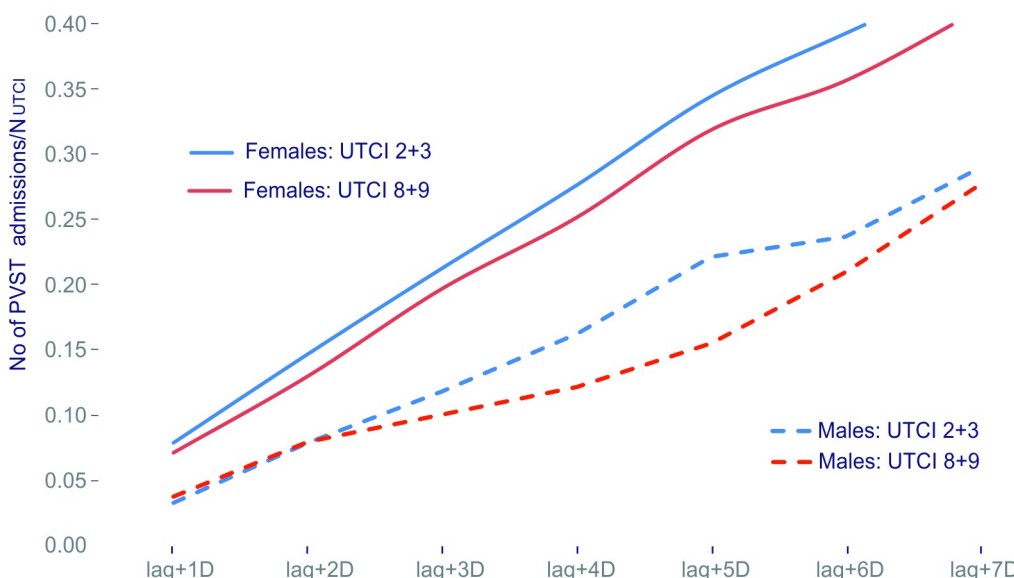

**Fig 5. Cumulative curves illustrating the number of patients with PSVT (females and males) since the occurrence of cold and hot stress.**

## Discussion

The occurrence of a PSVT episode is constrained by some boundary conditions, that is there must exist a specific pathology that predisposes a patient for the disease. Three critical factors are required: (1) the presence of an auxiliary circuit tract that facilitates the re-entry mechanism, (2) various electrophysiological properties of the pathways and (3) development of unidirectional block. This mechanism was proposed first by Ralph Mines in laboratory model and confirmed by Mark E. Josephson in humans [29, 30]. The second point (2) may be determined by structural changes (fibrosis) as well as external factors sympathetic and parasympathetic tone, hormonal changes and, more generally, anything that can cause homeostatic imbalance. The trigger of PSVT episode may be any stressor that activates adrenergic stimulation as long as the conditions are favorable, for example, hyperthyroidism, infection, dehydration [31–33]. The number of potential stressors is quite large–ranging from emotional stress associated with events that affect the entire population (elections, earthquake) to stress that occurs in everyday situations (anger, sexual activity) [32, 34].

Epidemiological analyses concerning the impact of biometeorological parameters on the occurrence of PTSV have not been published thus far. In the only related study that we have managed to find, the authors analyse the impact of atmospheric pressure, relative humidity, ambient temperature, and wind speed; however, the study group is relatively small (n = 362), with a history of cardiac diseases, and the analysed period is very short (January-April 2001) [35]. Interestingly, Čulić V et al. (we too) did not reveal statistically significant correlations for, apparently, such an obvious parameter as ambient temperature. On the other hand, the majority of reported by Čulić V et al. PSVT episodes occurred during daytime, thus during the time when the actual possibility of the impact of heat and cold stress triggered by atmospheric conditions exists. Considering the conditions of an Mediterranean country, it may be safely assumed that the impact of cold/heat stress during the night rest is slight as regards population values (only selected professions might be affected), whereas the impact of, for example, atmospheric pressure and relative humidity is independent of the time of the day, but our results come from East-European country and are not limited to season or day time.

In our results, cold/heat stress (Table 3) definitely triggers the cascade of the events, but only among females. In a healthy person, the physiological reaction that involves the stimulation of the thermoregulation center with the activation of the sympathetic nervous system leading to the acceleration of heart rate, the centralization of circulation and the increase in blood pressure, under "favorable" conditions, may trigger a pathological cascade of events. The increase in blood pressure in the aforementioned mechanism has been reported in a number of population-based studies [36, 37]. In this context, the report by Modesti PA et al. is very telling. Based on a 24-hour monitoring of blood pressure, a greater increase of blood pressure was recorded during the daytime on those days with lower ambient temperature [38].

The cause of paroxysmal supraventricular tachycardia is the formation of a circulating excitation wave (reentry). However, tachycardia can also be caused by increased or pathological cellular automatism or triggered activity [39]. Sex differences in the heart's electrical activity have already been proven in the literature [40]. In Table 3, we presented an increased risk of PSVT under heat stress conditions among females. We may hypothesize that: the increased activity of the parasympathetic nervous system that occurs during heat stress, the blood vessel dilating, and slower heart rate leads to a decrease in blood pressure. As a result of lowering blood pressure in the heart, microfocal of ischemia may occur, or additional conduction pathways may be activated, which favors the development of PSVT. Women have a resting heart rate of 3–5 beats/min higher than men [41]. Moreover, in women, there is an increased automaticity of the sinoatrial node, independent of the influence of the autonomic nervous system, and a shorter refractory time, contributing to the increased risk of PSVT in heat stress [42]. More difficult to explain is the dependence presented in Table 3 concerning men, suggesting the protective effect of cold. Supraventricular tachycardias come from various places of the heart, which are determined, among others, by sex [43].

In the available, extensive literature describing sex differences in response to cold and heat stress, there are reports of a greater reduction in heart rate among men than women during exposure to cold [44]. The problem here is the fact that most of these studies are based on clinical experiments on selected groups of people (military, athletes, divers) [45, 46]. Meanwhile, in the general population, we are dealing not only with phenomena based on physiological reactions, but also with a secondary effect covering a number of different components. For example, it is conceivable that men who are more frequently addicted to nicotine are less likely to go outside for a cigarette under severe cold stress, or have a lower need to look attractive, preferring warmer clothing [47, 48]. Thus, we cannot rule out the possibility that the weather-related results obtained in our study are secondary to the lifestyle.

On the other hand, in our results we observe a statistically significant ($p<0.05$) decrease in the amount of PSVT in the month of April among women (Fig 2) and it is noticeable in all (except the oldest) age groups (Fig 3). In terms of thermal conditions, April is similar (in Dfb climatic zone) to October, when we observed a relatively large number of PSVT episodes among women. However, despite the fact that both in spring and autumn there is a change in the clothing model, it should be remembered that outdoor clothing adjustment is influenced by the local past temperatures [49]. Ladies, who are more sensitive to heat/cold stress, better tolerate the increase than decrease in outdoor temperature [50].

Moreover, the perimenopausal group of females present lower thermal requirements [51]. In this life period, women experience a number of vasomotor symptoms that, as is already known, are better tolerated in adverse–cold–weather conditions [52, 53]. Cymes et al., who analysed the number of fainting events in correlation with the UTCI, and Skutecki et al., who examined the number of consultations within the healthcare system owing to hypertension, observe that the relatively least sensitive group to atmospheric conditions is this age group of females [37, 54]. Had we excluded from our analysis the perimenopause age group of females,

the RR of PSVT for females would have been 1.23 for cold stress and as high as 1.31 for heat stress (Table 3, Fig 4). In the EHRA (European Heart Rhythm Association) consensus, Linde C et al. report a higher incidence of PSVT in the luteal phase of the menstrual cycle in females [55]. In this document, it is also noted that, despite a wider range of PSVT symptoms, females are diagnosed later than males. This is most likely a consequence of the tendency to assign the psychogenic background of the condition as regards females [56]. Our study does not give grounds for far-reaching interpretations; however, we believe that this is rather the case of masking PSVT symptoms by hot flushes (experienced by nearly 80% of females of that age) than, for example, a consequence of a decrease in progesterone levels [57]. It should be remembered that the sense of thermal comfort in this group is shifted towards cooler ambient conditions [51].

Considering the problem from a broader perspective, the sensation of thermoneutral conditions depends on, for example, ambient temperature, but the range of such conditions is associated with environmental adaptation, and this–in long-term–results from evolutionary processes [58, 59]. Presently climate change is progressing much faster than before; the risk of extreme weather phenomena is increasing, as well as their consequences [60]. It is necessary to monitor their impact on the population health, and to identify groups of particularly vulnerable individuals [61–66]. It is worth noting that despite the data collected in fairly uniform climatic conditions, typical of Central Europe (Fig 2, Table 2) with the lack of seasonality of PSVT occurrence (Fig 3), the use of UTCI allowed to find important and statistically significant relationships for weather extremes. Our results indicate a greater risk of PSVT in strong and very strong cold and heat stress conditions, while identifying a group in which this condition may occur particularly frequently–females except for the group in the perimenopause stage (age group of 46–55).

The UTCI is ever more often employed in such analyses. It has ceased to be a biometeorological curiosity and is becoming one of the few basic tools that describe the potential impact of the atmospheric environment on the human organism [10, 16, 67]. The World Meteorological Organization (WMO) recommends including this index in the product portfolio offered by the National Meteorological Services [68, 69]. Our study extends the knowledge about the relationship between climate and humans. The results demonstrate that the use of an appropriate tool, in this case the comprehensive biometeorological index (UTCI), can help to find relationships with PVST that have not been described previously. From a medical point of view, we have proven the hypothesis that an increased incidence of PSVT occurs after extreme thermal episodes. The relationship between PSVT and meteorological conditions provides another argument for the promotion of the UTCI as a very flexible and useful meteorological tool for epidemiological analyses.

## Limitations

Since our study is based on the retrospective analysis of admissions to the healthcare system and because during the study period significant disruptions in the functioning of this system occurred (the COVID-19 epidemic), we compared the RR indicated in Table 3, dividing the analysed period into two intervals–before and during the pandemic. The COVID-19 epidemic is a recognized factor that interferes with the results of retrospective epidemiological analyses. In particular, initially, the pandemic significantly impacted on the number of patients seeking consultations within the healthcare system [70]. The absence of the COVID-19 epidemic impact on our results, in our view, results from a subjectively great (in the patient's opinion) sense of health hazard. A patient with supraventricular tachycardia that lasted long enough to induce him or her to seek assistance in acute care units and then being referred to the hospital

ED must have had a whole range of symptoms. This is one of the differences between our study and the one conducted by Čulić V et al [35]. In our study, by definition, there are no asymptomatic short PSVT episodes.

Some secondary dependencies might be the limitation of our study. UTCI values present seasonal fluctuations. Other parameters, like air or noise pollution, that change seasonally, are not considered in the UTCI structure [71].

Other limitations are typical of studies based on registers of medical diagnoses made during routine medical practice and may include single cases of false data, whereas their retrospective verification is practically impossible as it would require a manual re-analysis of the correctness of diagnoses. However, the size, period, and quantity of the analysed data allow us to marginalize this problem.

## Conclusions

Our cohort study conducted in 2016–2021 in three Polish cities: Olsztyn, Radom and Wroclaw, representing different bioclimatic conditions, showed that unfavorable biometeorological conditions, defined as heat or cold stress, expressed by the Universal Climate Thermal Index may be a risk factor for paroxysmal supraventricular tachycardia. We found the increased risk of PSVT in women during heat/cold stress in comparison to thermoneutral conditions. Females, particularly at the perimenopausal stage and men irrespective of age were less likely to develop PSVT under thermal stress as compared to thermoneutral conditions. The higher risk was found in females, with the exception of the perimenopausal group, under both cold stress (RR = 1.74, p< 0.001) and heat stress (RR = 1.56, p = 0.029) conditions. Progress in climate change requires searching for universal methods and tools to monitor relationships between humans and climate. Our paper confirms that the UTCI is the universal tool describing the impact of thermal stress on the human body and its high usefulness in medical researches.

## Supporting information

**S1 Table. Database used in the study.** –on reasonable request.
(XLSX)

**S2 Table. Relative risk ratios.**
(XLSX)

**S1 Graphical abstract.**
(TIF)

## Author Contributions

**Conceptualization:** Rakesh Jalali, Jerzy Romaszko.

**Data curation:** Rakesh Jalali, Ewa Dragańska, Leszek Gromadziński, Iwona Cymes, Janusz Bernard Sokołowski, Magdalena Poterała, Leszek Markuszewski.

**Investigation:** Iwona Cymes, Katarzyna Glińska-Lewczuk.

**Methodology:** Jerzy Romaszko.

**Project administration:** Rakesh Jalali, Jerzy Romaszko.

**Supervision:** Katarzyna Glińska-Lewczuk.

**Visualization:** Anna Jeznach-Steinhagen.

**Writing – original draft:** Rakesh Jalali, Jerzy Romaszko, Anna Maria Romaszko-Wojtowicz, Katarzyna Glińska-Lewczuk.

**Writing – review & editing:** Rakesh Jalali, Jerzy Romaszko, Ewa Dragańska, Leszek Gromadziński, Iwona Cymes, Janusz Bernard Sokołowski, Magdalena Poterała, Leszek Markuszewski, Anna Maria Romaszko-Wojtowicz, Anna Jeznach-Steinhagen, Katarzyna Glińska-Lewczuk.

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
