## [Decision Letter · Decision Letter 0]

21 Nov 2023

PONE-D-23-27236Heat and cold stress increases the risk of paroxysmal supraventricular tachycardiaPLOS ONE

Dear Dr. Romaszko,

Thank you for submitting your manuscript to PLOS ONE. After careful consideration, we feel that it has merit but does not fully meet PLOS ONE’s publication criteria as it currently stands. Therefore, we invite you to submit a revised version of the manuscript that addresses the points raised during the review process.

We look forward to receiving your revised manuscript.

Kind regards,

Hidenori Otani, Ph.D.

Academic Editor

PLOS ONE

Journal Requirements:

lease ensure that your manuscript meets PLOS ONE's style requirements, including those for file naming. The PLOS ONE style templates can be found at https://journals.plos.org/plosone/s/file?id=wjVg/PLOSOne_formatting_sample_main_body.pdf and https://journals.plos.org/plosone/s/file?id=ba62/PLOSOne_formatting_sample_title_authors_affiliations.pdf 2. Did you know that depositing data in a repository is associated with up to a 25% citation advantage (https://doi.org/10.1371/journal.pone.0230416)? If you’ve not already done so, consider depositing your raw data in a repository to ensure your work is read, appreciated and cited by the largest possible audience. You’ll also earn an Accessible Data icon on your published paper if you deposit your data in any participating repository (https://plos.org/open-science/open-data/#accessible-data). 3. We note that Figure 1 in your submission contain map images which may be copyrighted. All PLOS content is published under the Creative Commons Attribution License (CC BY 4.0), which means that the manuscript, images, and Supporting Information files will be freely available online, and any third party is permitted to access, download, copy, distribute, and use these materials in any way, even commercially, with proper attribution. For these reasons, we cannot publish previously copyrighted maps or satellite images created using proprietary data, such as Google software (Google Maps, Street View, and Earth). For more information, see our copyright guidelines: http://journals.plos.org/plosone/s/licenses-and-copyright. We require you to either present written permission from the copyright holder to publish these figures specifically under the CC BY 4.0 license, or (2) remove the figures from your submission: a. You may seek permission from the original copyright holder of Figure 1 to publish the content specifically under the CC BY 4.0 license.   We recommend that you contact the original copyright holder with the Content Permission Form (http://journals.plos.org/plosone/s/file?id=7c09/content-permission-form.pdf) and the following text:“I request permission for the open-access journal PLOS ONE to publish XXX under the Creative Commons Attribution License (CCAL) CC BY 4.0 (http://creativecommons.org/licenses/by/4.0/). Please be aware that this license allows unrestricted use and distribution, even commercially, by third parties. Please reply and provide explicit written permission to publish XXX under a CC BY license and complete the attached form.” Please upload the completed Content Permission Form or other proof of granted permissions as an "Other" file with your submission. In the figure caption of the copyrighted figure, please include the following text: “Reprinted from [ref] under a CC BY license, with permission from [name of publisher], original copyright [original copyright year].” b. If you are unable to obtain permission from the original copyright holder to publish these figures under the CC BY 4.0 license or if the copyright holder’s requirements are incompatible with the CC BY 4.0 license, please either i) remove the figure or ii) supply a replacement figure that complies with the CC BY 4.0 license. Please check copyright information on all replacement figures and update the figure caption with source information. If applicable, please specify in the figure caption text when a figure is similar but not identical to the original image and is therefore for illustrative purposes only.The following resources for replacing copyrighted map figures may be helpful: USGS National Map Viewer (public domain): http://viewer.nationalmap.gov/viewer/The Gateway to Astronaut Photography of Earth (public domain): http://eol.jsc.nasa.gov/sseop/clickmap/Maps at the CIA (public domain): https://www.cia.gov/library/publications/the-world-factbook/index.html and https://www.cia.gov/library/publications/cia-maps-publications/index.htmlNASA Earth Observatory (public domain): http://earthobservatory.nasa.gov/Landsat: http://landsat.visibleearth.nasa.gov/USGS EROS (Earth Resources Observatory and Science (EROS) Center) (public domain): http://eros.usgs.gov/#Natural Earth (public domain): http://www.naturalearthdata.com/ 4. Please include captions for your Supporting Information files at the end of your manuscript, and update any in-text citations to match accordingly. Please see our Supporting Information guidelines for more information: http://journals.plos.org/plosone/s/supporting-information. 
Please review your reference list to ensure that it is complete and correct. If you have cited papers that have been retracted, please include the rationale for doing so in the manuscript text, or remove these references and replace them with relevant current references. Any changes to the reference list should be mentioned in the rebuttal letter that accompanies your revised manuscript. If you need to cite a retracted article, indicate the article’s retracted status in the References list and also include a citation and full reference for the retraction notice.

Reviewers' comments:

Reviewer's Responses to Questions

**Comments to the Author**

1. Is the manuscript technically sound, and do the data support the conclusions?

Reviewer #1: Partly

Reviewer #2: Yes

Reviewer #3: Yes

2. Has the statistical analysis been performed appropriately and rigorously? 

Reviewer #1: I Don't Know

Reviewer #2: Yes

Reviewer #3: Yes

3. Have the authors made all data underlying the findings in their manuscript fully available?

Reviewer #1: No

Reviewer #2: Yes

Reviewer #3: Yes

4. Is the manuscript presented in an intelligible fashion and written in standard English?

Reviewer #1: Yes

Reviewer #2: Yes

Reviewer #3: Yes

5. Review Comments to the Author

Reviewer #1: Although the role of climate in diseases is undoubtedly important, I consider that in such a large study many other variables need to be assessed that are impossible to do due to the nature of the study.

Reviewer #2: This article examine the effect of heat and cold stresses on paroxysmal supraventricular

tachycardia. The authors conclude that UTCI is the universal tool describing the impact of thermal stress on the human body and its high usefulness in medical researches. This paper is labor intensive and it is well written. I have no further comments. I commend the authors for their time and effort in this study.

Minor comments: for table 1 and 3, maybe put down years below the age?

Reviewer #3: the gaps from previous research, then justify your research to explain the innovations made. then add the purpose of making this research and explain what contribution is obtained to the public, add justification and contribution to the last paragraph

6. PLOS authors have the option to publish the peer review history of their article (what does this mean?). If published, this will include your full peer review and any attached files.

Reviewer #1: No

Reviewer #2: **Yes: **Tze-Huan Lei

Reviewer #3: No

---

## [Author Response · Author response to Decision Letter 0]

26 Nov 2023

Response to Reviewers' and Editors’ comments

We want to thank the Reviewers for their insightful reviews of our manuscript. Please, find below the Reviewers’ comments written in bold and our responses inserted after each of their comments. The responses refer to the line numbering from the original (first) version. Next, our response to editorial issues is prepared in this same way.

Reviewer 1

Although the role of climate in diseases is undoubtedly important, I consider that in such a large study many other variables need to be assessed that are impossible to do due to the nature of the study.

AD. We treat this remark as a comment rather than a suggestion to correct the text. We share the reviewer's opinion, this type of research always has a certain risk of secondary dependencies. We write about it in the limitations section. We agree with the reviewer that climatic factors are among the increasingly important determinants of many diseases. Therefore, we would like to emphasize the applicability of UTCI when climate-related diseases are of concern.

Reviewer 2

AD. Reviewer 2 does not raise any critical comments. We want to thank him for his positive attitude to our article. 

Minor comments: for table 1 and 3, maybe put down years below the age?

AD. Corrected according to suggestion and consequently Fig. 3 as well.

Reviewer 3

The gaps from previous research, then justify your research to explain the innovations made. then add the purpose of making this research and explain what contribution is obtained to the public, add justification and contribution to the last paragraph.

AD. The last paragraph of the discussion section has been supplemented with the relevant explanations as suggested by the Reviewer.

“Our study extends the knowledge about the relationship between climate and humans. The results demonstrate that the use of an appropriate tool, in this case the comprehensive biometeorological index (UTCI), can help to find relationships with PVST that have not been described previously. From a medical point of view, we have proven the hypothesis that an increased incidence of PSVT occurs more often after extreme thermal episodes.”

Editors’ comments

1. When submitting your revision, we need you to address these additional requirements. Please ensure that your manuscript meets PLOS ONE's style requirements, including those for file naming.

AD. The text has been formatted according to the Journal requirements.

2. Did you know that depositing data in a repository is associated with up to a 25% citation advantage…

AD. Our database is available as S1 Table in supplementary material 

3. We note that Figure 1 in your submission contain map images which may be copyrighted. All PLOS content is published under the Creative Commons Attribution License (CC BY 4.0), which means that the manuscript, images, and Supporting Information files will be freely available online, and any third party is permitted to access, download, copy, distribute, and use these materials in any way, even commercially, with proper attribution. For these reasons, we cannot publish previously copyrighted maps or satellite images created using proprietary data, such as Google software (Google Maps, Street View, and Earth).

AD. The graphics (including the borders of Europe) in Figure 1 have been adapted and modified from the authors' private, unpublished material. Figure 1 was created by the authors in CorelDraw based on the data collected during the study. It has never been published before.

4. Please include captions for your Supporting Information files at the end of your manuscript, and update any in-text citations to match accordingly.

AD. Done.

We remain at your disposal.

On behalf of all authors,

Jerzy Romaszko

---

## [Decision Letter · Decision Letter 1]

12 Dec 2023

Heat and cold stress increases the risk of paroxysmal supraventricular tachycardia

PONE-D-23-27236R1

Dear Dr. Romaszko,

We’re pleased to inform you that your manuscript has been judged scientifically suitable for publication and will be formally accepted for publication once it meets all outstanding technical requirements.

Kind regards,

Hidenori Otani, Ph.D.

Academic Editor

PLOS ONE

Additional Editor Comments (optional):

Reviewers' comments:

Reviewer's Responses to Questions

**Comments to the Author**

1. If the authors have adequately addressed your comments raised in a previous round of review and you feel that this manuscript is now acceptable for publication, you may indicate that here to bypass the “Comments to the Author” section, enter your conflict of interest statement in the “Confidential to Editor” section, and submit your "Accept" recommendation.

Reviewer #2: All comments have been addressed

2. Is the manuscript technically sound, and do the data support the conclusions?

Reviewer #2: Yes

3. Has the statistical analysis been performed appropriately and rigorously? 

Reviewer #2: Yes

4. Have the authors made all data underlying the findings in their manuscript fully available?

Reviewer #2: Yes

5. Is the manuscript presented in an intelligible fashion and written in standard English?

Reviewer #2: Yes

6. Review Comments to the Author

Reviewer #2: All comments are satisfactory addressed. Well done to the authors for time time and effort into this study!

7. PLOS authors have the option to publish the peer review history of their article (what does this mean?). If published, this will include your full peer review and any attached files.

Reviewer #2: **Yes: **Tze-Huan Lei

---

## [Editor Report · Acceptance letter]

18 Dec 2023

PONE-D-23-27236R1 

PLOS ONE

Dear Dr. Romaszko, 

I'm pleased to inform you that your manuscript has been deemed suitable for publication in PLOS ONE. Congratulations! Your manuscript is now being handed over to our production team.

Kind regards, 

on behalf of

Dr. Hidenori Otani 

Academic Editor

PLOS ONE